# Green Infrastructure and Water: An Analysis of Global Research

José Luis Caparrós-Martínez, Juan Milán-García, Nuria Rueda-López * and Jaime de Pablo-Valenciano

Department of Business and Economics, University of Almeria, 04120 Almeria, Spain;
jlcaparrosm@gmail.com (J.L.C.-M.); jmg483@ual.es (J.M.-G.); jdepablo@ual.es (J.d.P.-V.)
* Correspondence: nrueda@ual.es

**Abstract:** Green infrastructure (GI) is a nature-based solution that encompasses all actions that rely on ecosystems and the services they provide to respond to various societal challenges such as climate change, food security or disaster risk. The objective of this work is to analyze the state of the art and latest trends in research on GI related to the water cycle for the period 2002–2019. For this purpose, a bibliometric study is carried out taking as reference the two most important scientific databases, Web of Science (WoS) and Scopus. The results show that, as of 2013, there is an exponential increase in the number of publications. This is due to the fact that significant regions of the planet, such as Europe, have adopted strategies aimed at promoting the use of GI since 2013. The keyword analysis points out that ecosystem services is the most relevant concept, which shows the capacity of these infrastructures to facilitate multiple goods and services related to the water cycle. New lines of research are opened up which are based on the analysis of other elements of GI related to water, such as groundwater.

**Keywords:** green infrastructure; water resources; environmental services; sustainability; nature-based solutions; bibliometric analysis

## 1. Introduction

Water is an element of nature that is essential for human health, well-being and development, as well as for the conservation of the planet's ecosystems and natural habitats [1–3]. In July 2010, the United Nations General Assembly recognized its importance, establishing that every human being has the right to between 50 and 100 liters of safe and affordable water per person per day, and that access should be within 1000 meters or a maximum of half an hour from home [4].

However, the extensive urbanization and the accelerated change in land uses that are taking place in different regions of the world are causing the over-exploitation and degradation of natural ecosystems, especially those related to the water cycle, such as rivers, aquifers or wetlands [5,6]. In this regard, it should be emphasized that agriculture currently represents about 70% of global freshwater use [7]. This damage and over-exploitation of water resources produces a series of environmental conditions such as the decrease in rainwater infiltration and aquifer recharge, the widespread loss of water quality, and an increase in the problems derived from floods and torrential rains [6,8]. The gradual deterioration of water resources together with the continued increase in worldwide consumption, 1% per year since the 1980s [9], has resulted in more than two billion people living in water-scarce countries; four billion people suffering from severe water shortages for at least one month a year; three out of ten people in the world without access to safe drinking water; and six out of ten people not having access to safe sanitation services [9–11]. Both the decrease in freshwater resources by 40%

and the increase in the world population for 2030, as foreseen by the United Nations World Water Assessment Programme (WWAP) [12] could generate a world water crisis.

If we add to this situation the potential expected effects of climate change, such as increases in catastrophic storms and long periods of drought, we can affirm that in the near future and in many regions the security and sustainability of water for local populations is at risk [6–8,13]. There are climate models that predict that by 2050 an increase of 1.5 °C in the average global temperature of the planet could cause drought and habitat degradation that would make life difficult for 178 million people around the world, while the effect on the population of an increase of between 2 °C and 3 °C would be significantly higher, affecting between 220 and 277 million people, respectively [7].On the other hand, the number of people at risk of flooding is expected to increase from 1.2 billion today to 1.6 billion in 2050 (about 20% of the world´s population), which will cause the economic value of assets at risk to be about $45 trillion by 2050, a growth of more than 340% compared to 2010 [14].

In view of this scenario of climate change, demographic growth, increased urbanization rates and intensification of pollution, and degradation of water resources, it is essential and urgent that different countries and regions promote the implementation of measures that help transform the way water is managed. To respond to these great challenges, the different regions of the world can adopt, on the one hand, engineering or technological strategies; and/or, on the other hand, an alternative approach based on comprehensively managing natural and social systems in order to increase the benefits that nature provides for both human well-being, health and development [15]. In the latter case, this refers to Nature-based Solutions (NbS), a concept defined by the International Union for Conservation of Nature (IUCN) that "encompasses all actions that rely on ecosystems and the services they provide, to respond to various societal challenges such as climate change, food security or disaster risk" [16].

For too long human-made infrastructure, so-called gray infrastructure, has been used to solve water problems. Consequently, NbS which focus their attention on GI based on using natural and semi-natural areas to provide alternative water resource management have been neglected [17,18]. The great attraction of GI lies in the fact that it offers important environmental services related to water from three fundamental perspectives: smart growth, climate change adaptation, and health and wellbeing (Figure 1).

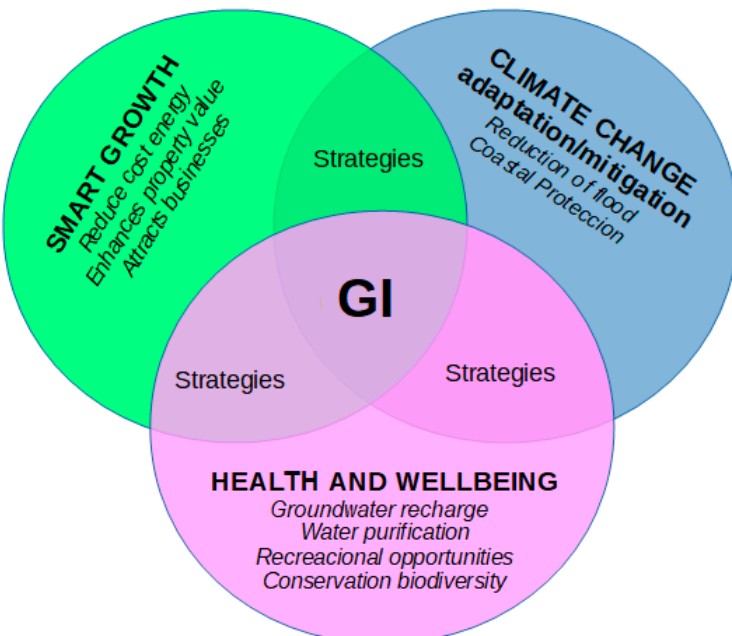

**Figure 1.** Environmental services provided by green infrastructure (GI) related to the water cycle. Source: Own elaboration.

Despite the fact that the term GI has recently become popularized, there is no universally accepted definition, as it is applied at different scales, for different issues, by different actors, including public managers, researchers and the general public [19].

Consequently, GI is defined by researchers in different ways (see Table 1), although there is general agreement that this term is multi-functional and delivers both ecological and social benefits [20]. Although water issues are usually included in the various definitions of GI, when processes and components related to water and aquatic systems become particularly relevant to the solution of management problems, some authors use the term Blue Infrastructures as an alternative approach to GI [21]. Within the framework of this concept, water bodies (lakes, lagoons, marshes, swamps), rivers, streams, springs and coastal ecosystems would be included.

**Table 1.** Definitions of GI.

| Author | Definition |
|---|---|
| [22] | "Green infrastructure is an engineered intervention that uses vegetation, soils, and natural processes to manage water and create healthier built environments for people and the natural resources that sustain them. GI can range in scale from small-scale technologies such as rain gardens and green roofs to regional planning strategies targeting conservation or restoration of natural landscapes and watersheds. GI approaches may be interconnected with existing and planned grey infrastructure networks to create sustainable infrastructure that can enhance community resilience to disasters and climate change as a result of increased water retention and groundwater recharge, flood mitigation, erosion control, shoreline stabilization, combatting urban heat island effect, improving water quality, conserving energy for buildings." |
| [23] | Urban GI relates to a planning approach aimed at developing networks of green and blue spaces in urban areas to deliver a wide range of ecosystem services. |
| [24] | A strategically planned network of natural and semi-natural areas with other environmental features designed and managed to deliver a wide range of ecosystem services in both rural and urban settings. The European Commission's definition of green infrastructure also incorporates "blue spaces" in reference to aquatic ecosystems, including coastal and marine ecosystems. |
| [25] | Green infrastructure is the network of natural and semi-natural areas, features and green spaces in rural and urban, terrestrial, freshwater, coastal and marine areas, which together enhance ecosystem health and resilience, contribute to biodiversity conservation and benefit human populations through the maintenance and enhancement of ecosystem services. Green infrastructure can be strengthened through strategic and coordinated initiatives that focus on maintaining, restoring, improving and connecting existing areas and features as well as creating new areas and features. |
| [26] | Green infrastructure is an approach to wet weather management that uses soils and vegetation to utilize, enhance and/or mimic the natural hydrologic cycle processes of infiltration, evapo-transpiration and reuse. |
| [27] | All natural, semi-natural and artificial networks of multifunctional ecological systems within, around and between urban areas, at all spatial scales. |
| [28] | Green infrastructure is a concept that is principally structured by a hybrid hydrological/drainage network, complementing and linking relict green areas with built infrastructure that provides ecological functions. |
| [29] | An interconnected network of natural areas and other open spaces that conserves natural ecosystem values and functions, sustains clean air and water, and provides a wide array of benefits to people and wildlife. |
| [30] | Green infrastructure comprises the provision of planned networks of linked multifunctional green spaces that contribute to protecting natural habitats and biodiversity, enable response to climate change and other biosphere changes, enable more sustainable and healthy lifestyles, enhance urban livability and wellbeing, improve the accessibility of key recreational and green assets, support the urban and rural economy and assist in the better long-term planning and management of green spaces and corridors. |

**Table 1.** *Cont.*

| Author | Definition |
| --- | --- |
| [31] | The abundance and distribution of natural features in the landscape like forests, wetlands, and streams. Just as built infrastructure like roads and utilities is necessary for modern societies, green infrastructure provides the ecosystem services that are equally necessary for our well-being. |
| [32] | Our nation's natural life support system - an interconnected network of protected land and water that supports native species, maintains natural ecological processes, sustains sir and water resources and contributes to the health and quality of life for America's communities and people. |
| [33] | An interconnected network of green space that conserves natural ecosystem values and functions and provides associated benefits to human populations. |

Source: Own elaboration.

Investment in GI is based on the logic that it will always be more profitable to invest in NbS than to replace these environmental services with human technological solutions [34]. The development and conservation of a region´s GI is considered a sound strategy for nature, the economy and employment with a series of advantages [21,35–37], among which the following are worth noting (Figure 2):

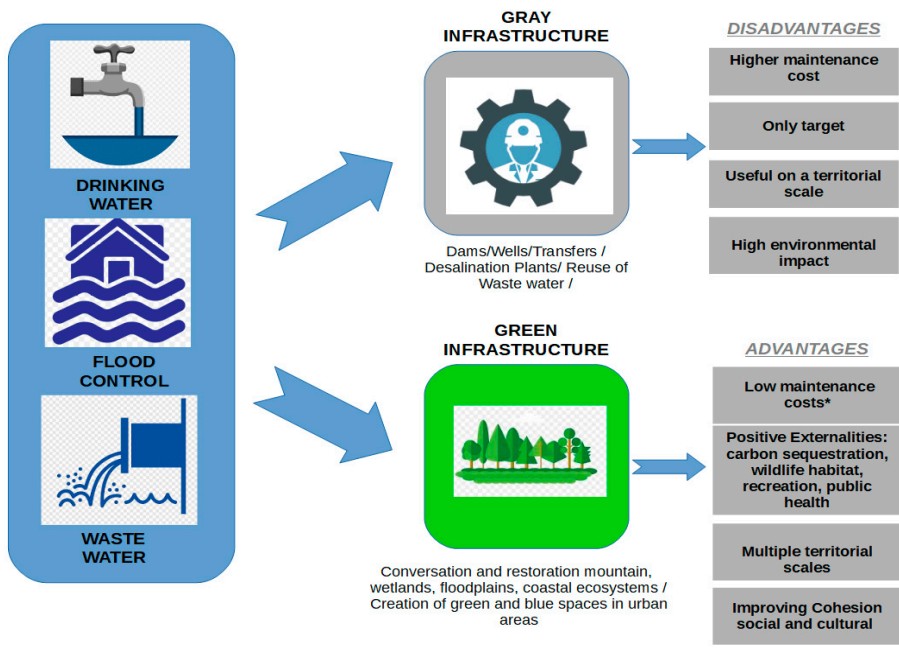

**Figure 2.** Gray vs. GI. Source: Own elaboration.

*Economic cost savings*. When GI is properly managed and maintained, it can lead to considerable cost savings, as it tends to improve the environmental services it provides over time, through the establishment of healthy ecosystems, the recovery of damaged habitats and the reconnection of natural and semi-natural fragmented areas. In contrast, gray infrastructure requires continuous investments to adapt to the rate of population growth and to cover its wear and tear [35]. A clear example of gray infrastructure is the control of coastal erosion through the restoration of dunes, compared to the alternative of using coastal levees and breakwaters that tend to degrade over time.

*Multifunctional character and lower environmental cost*. Unlike gray infrastructure, which usually has a single objective, investments in water-related GI usually produce other positive externalities. Thus, for example, a treatment plant only serves to purify wastewater, while a natural wetland, in addition to purifying water, conserves biodiversity and is a carbon sink, among other advantages. On the other

hand, investments in the conservation and improvement of the forests of receiving basins, in addition to improving the filtration and natural recharge of aquifers, also have other advantages, including help in controlling erosion, generating greater carbon sequestration, improving habitats and creating new recreational opportunities for the population. However, for this to happen, the ecosystem must be healthy [35,37,38].

This multifunctionality of the GIs, which allows synergy and compensation between different environmental services, not only includes ecological aspects, but also includes important social aspects, especially in urban areas, which are important to identify, such as sustainable development, environmental justice and social cohesion [39,40].

*Ability to adapt to different territorial scales.* Unlike gray infrastructure, which normally operates locally, GI is close to a fractal structure, with elements ranging from the continental scale (for example, a large transnational mountain range that functions as an ecological corridor), to elements of smaller dimensions with value for the provision of environmental services at the local or urban level (for example, a wetland, which is responsible for purifying water and protecting against possible floods).

In this sense, if the interactions between ecological processes and human activities taking place at the local scale in the GIs of urban landscapes and socio-ecological systems are properly planned and managed, GIs will contribute to the conservation of biodiversity, the improvement of environmental quality, the reduction of the ecological footprint of urban environments, the adaptation of cities to climate change, social cohesion through the provision of spaces for social interaction, the alleviation of stress and fatigue and the promotion of volunteerism [27,41–43].

GI is configured as a very effective management tool, since from this approach alternatives can be proposed at different territorial levels.

Despite these advantages of GI, this type of action is frequently neglected by managers when making decisions or planning investments. This is largely due to the fact that, unlike what happens in gray infrastructure, the evidence of the benefits of GI is more difficult to quantify than the costs associated with its implementation [24]. In general, this type of action requires more time for the results to be visible and it is difficult to assess their contribution in market terms. A reason for this is that in most of the territories there are gaps regarding the availability of historical data and economic valuation tools for environmental services [38]. Despite this, studies carried out to date in different areas worldwide show greater profitability in the alternatives proposed from the GI approach than those proposed from the perspective of gray infrastructure [37]. There is also scientific evidence that the gradual loss or degradation of the GIs that exist in or around urban areas causes long-term economic losses and affects the social and cultural values of cities [44]. Furthermore, epidemiological studies show a positive relationship between longevity, health and access to elements of urban GIs [45,46].

Although it is considered to be in the early stages of development, various studies have shown the effectiveness and benefits of using GI for stormwater management, improving water quality, or retaining runoff [6,47,48]. Regarding water resources, GI is presented as an important instrument for achieving and maintaining the health of aquatic ecosystems and offers multiple benefits related to increasing the availability of water for different uses, water purification, and conservation and protection of aquatic biodiversity, as well as for the adaptation and mitigation of the effects of climate change, such as floods, torrential rains or long periods of drought [49].

On the other hand, in the scientific field, the number of publications focused on environmental services provided by natural ecosystems has increased considerably in the last three decades. Countries in Europe and North America have been the first to delve deeply and investigate these issues. Although it started publishing later, China has also rapidly promoted this line of research [6,50,51].

For the above reasons, it is of great interest to understand the evolution of the publications that relate to these two concepts, GI and the water cycle, as well as the main areas of knowledge in which they have been developed. The knowledge generated by these studies is a very useful tool for the environmental policy agenda. The aim of this bibliometric study is to analyze the evolution of the scientific literature related to GI and the water cycle, that is, not only to analyze the current state of

research on this specific topic but also identify trends and future lines of research. Bibliometric analysis permits the researcher to evaluate developments in knowledge on a specific subject and assesses the scientific influence of researchers and sources [52]. There are already systematic reviews of the literature based on GI [53] or related to a specific type of GI, such as greenways [54], meta-analysis [55] and other type of literature reviews [27]. There are, as well, bibliometric analyses about water quality [56], conservation [57–59] or irrigation [60]. However, to the best of our knowledge, there are no studies in the specialized literature that link the concepts of GI and water from a bibliometric perspective. In fact, the main contribution of this work is that it presents the current state of knowledge about GI that serves as a basis for the study, design and implementation of new water management models for both scientists and policy-makers.

## 2. Materials and Methods

In order to analyze the link between GI and water, a bibliometric analysis has been performed. It is a method that mixes statistical and mathematical techniques to analyze research results [61,62], which is widely accepted by leading research institutions [63]. According to the Organisation for Economic Co-operation and Development (OECD) [64], it is defined as "the statistical analysis of books, articles, or other publications to measure the output of individuals/research teams, institutions, and countries, to identify national and international networks, and to map the development of new (multi-disciplinary) fields of science and technology". This work also uses the h-index to explain the performance or production of a research work. This is defined as the number "x" of articles with a total number of citations ≥ "x", so that those articles have been cited at least "x" times [65].

A series of steps have been followed in the bibliometric analysis (Figure 3). First, the search criteria, the keywords and the study period were defined. In this article, we have chosen to use the words "green infrastructure" and "water". These terms have been selected in order to obtain the most consistent results in relation to the central theme of this study, that is, a scientific mapping which demonstrates the importance of GI for the sustainable management of water resources. Scopus and Web od Science (WoS) were the databases selected to perform the analysis, since they are the two most relevant data sources given the rigorous protocol to which they adhere that ensures that the articles they include have a high level of quality [66]. The period analyzed runs from the year in which the first article on this subject is registered in each database (2002 in the case of WoS and 2005 in the case of Scopus) until 2019.

With regards to articles, books, conference proceedings and other research documents included in the main collection of WoS and Scopus, the number of documents added up to 891 and 1249 publications, respectively (Table 2). Similarly to the selection of the databases mentioned above, both results have been filtered to only include articles, thus guaranteeing the quality of the research, as these types of documents must undergo a process of review (thus excluding proceedings, reviews, books and book chapters). Furthermore, by type of document, research articles represent a high percentage of the distribution of the scientific contribution. Therefore, the exclusive analysis of articles reflects the state of the art of the importance of GI in water management. Finally, the main results were codified and analyzed, identifying common and divergent elements between both databases with respect to the fields of GI and water.

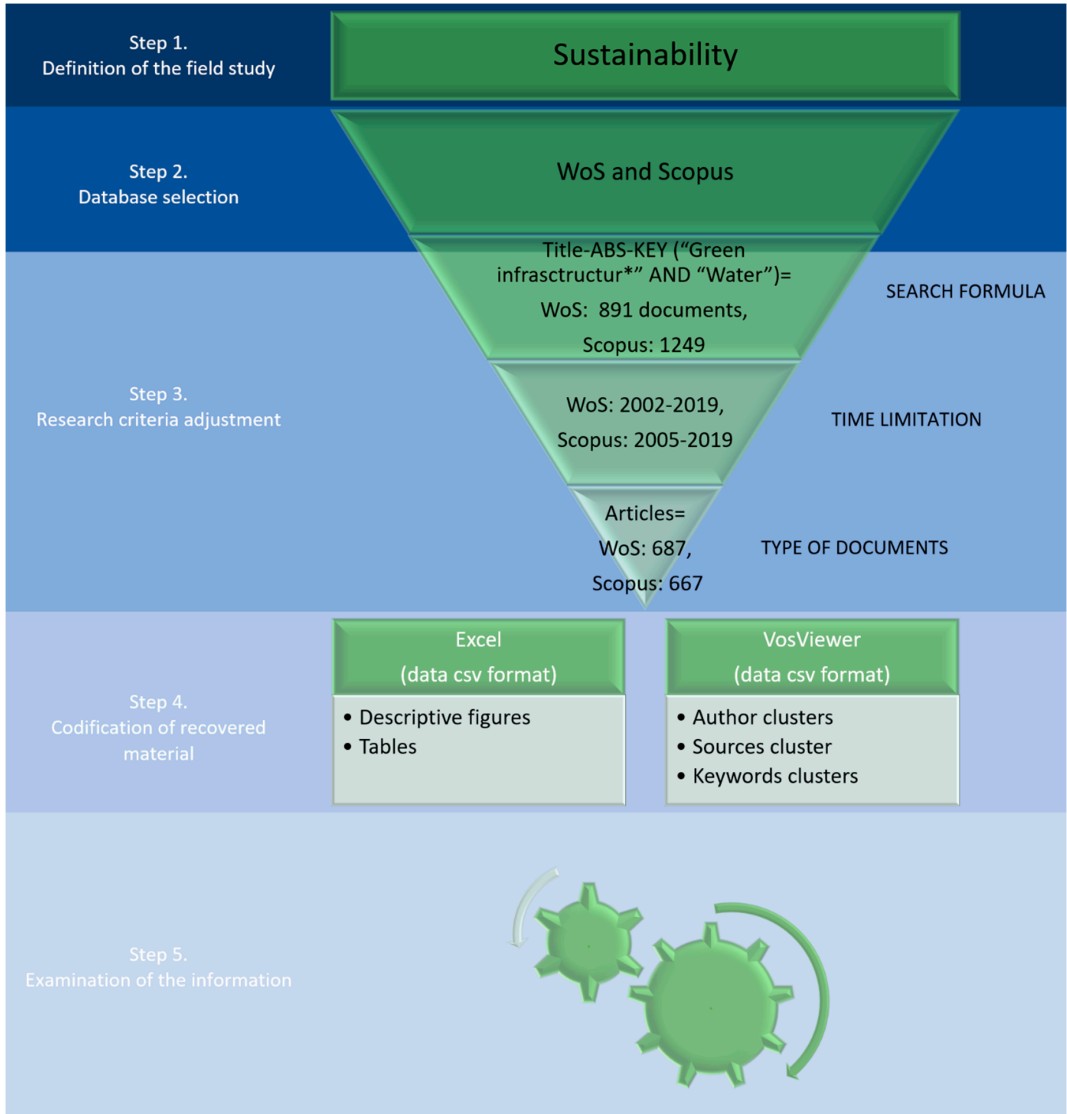

**Figure 3.** Data Gathering Process. Source: Own elaboration.

**Table 2.** Distribution of publications by type of document.

| Type of Document | WoS | Scopus |
|---|---|---|
| Article | 687 | 667 |
| Proceedings | 114 | 405 |
| Review | 62 | 97 |
| Book and Book Chapter | 28 | 80 |

Source: Own elaboration with Web of Science (WoS) and Scopus data (2019).

## 3. Results and Discussion

### 3.1. Number of Publications per Year

Research on GI and water began in the 21st century in both databases, which shows that the term is relatively novel, and is now being widely discussed among scientific circles. Specifically, the first article registered in the WoS database corresponds to the study by Marsalek & Chocat [67] entitled "International report: Stormwater management" in which stormwater management is analyzed

through expert surveys. However, in Scopus, the first work identified is that of Scholl & Schwartz [68] entitled "Making your resources count", in which they analyze the importance of natural resources for economies based on activities in the service sector (Table 3).

**Table 3.** Annual Distribution of Publications.

| | **WoS** | | | | **Scopus** | | | |
|---|---|---|---|---|---|---|---|---|
| **Year** | **A** | **TC** | **TC/A** | **H-Index** | **A** | **TC** | **TC/A** | **H-Index** |
| 2002 | 1 | 63 | 63 | 1 | - | - | - | - |
| 2005 | - | - | - | - | 1 | 1 | 1 | 1 |
| 2007 | 1 | 11 | 11 | 1 | 3 | 17 | 5.67 | 1 |
| 2008 | 3 | 276 | 92 | 3 | 5 | 348 | 69.6 | 4 |
| 2009 | 4 | 118 | 29.5 | 4 | 3 | 52 | 17.33 | 2 |
| 2010 | 8 | 247 | 30.9 | 5 | 12 | 284 | 23.67 | 6 |
| 2011 | 8 | 503 | 62.9 | 5 | 8 | 295 | 36.87 | 5 |
| 2012 | 16 | 331 | 20.7 | 9 | 13 | 305 | 23.46 | 8 |
| 2013 | 37 | 1541 | 41.6 | 20 | 39 | 1336 | 34.26 | 22 |
| 2014 | 33 | 630 | 19.1 | 15 | 39 | 691 | 17.72 | 17 |
| 2015 | 50 | 721 | 14.4 | 16 | 54 | 1208 | 22.37 | 19 |
| 2016 | 77 | 1138 | 14.8 | 21 | 84 | 1500 | 17.86 | 24 |
| 2017 | 122 | 1388 | 11.4 | 19 | 97 | 1528 | 15.75 | 20 |
| 2018 | 154 | 896 | 5.82 | 14 | 148 | 972 | 6.57 | 14 |
| 2019 | 173 | 282 | 1.63 | 7 | 161 | 261 | 1.62 | 8 |

Source: Own elaboration with Web of Science and Scopus data (2019). Notes: Y: Year; A: Articles; TC: Total Cites; H: H-index.

Since the publication of these studies, the number of articles has increased steadily and regularly over time. Until 2016, Scopus generally includes a higher number of articles (except in 2009, 2011 and 2012) than WoS. From this point on, this trend is reversed with WoS registering the largest number of articles in this area (Figure 4). In addition, it is observed that from 2013 there is an exponential increase in the number of publications in both databases. More precisely, it is in 2013 when, in different regions such as Europe, strategies are adopted aimed at promoting the use of GI [49] by recognizing that it is one of the main tools for addressing threats to biodiversity and for developing NbS. In addition, trends indicate that the number of publications on GI and water cycle is growing steadily, particularly over the last three years in both databases.

In contrast, the evolution in the number of citations does not present as regular a trend as the number of articles (Figure 5). Throughout the study period, several ups and downs are observed, also reaching the maximum value in 2013 in both databases for the same reason discussed above in the case of the number of publications.

Likewise, the most cited articles (Table 4) correspond to the studies by Gomez-Baggethun & Barton [69], with 483 citations, who value ecosystem restoration and conservation services when configuring urban planning; and Pataki et al. [70] with 386 citations, who follow the previous theme with regards to the incorporation of green solutions. These articles are followed by the studies by Barthel & Isendahl [71], with 143 citations, in which the influence of urban gardens, agriculture and water management on long-term agri-food security is analyzed; that of Coutts et al. [72], with 130 citations, where the existing literature is reviewed in order to demonstrate the potential of water-sensitive urban design to help improve thermal comfort in urban areas; and that of Lee et al. [73], also with 130 citations, who

study the integrated rainwater analysis and treatment system developed by the US Environmental Protection Agency.

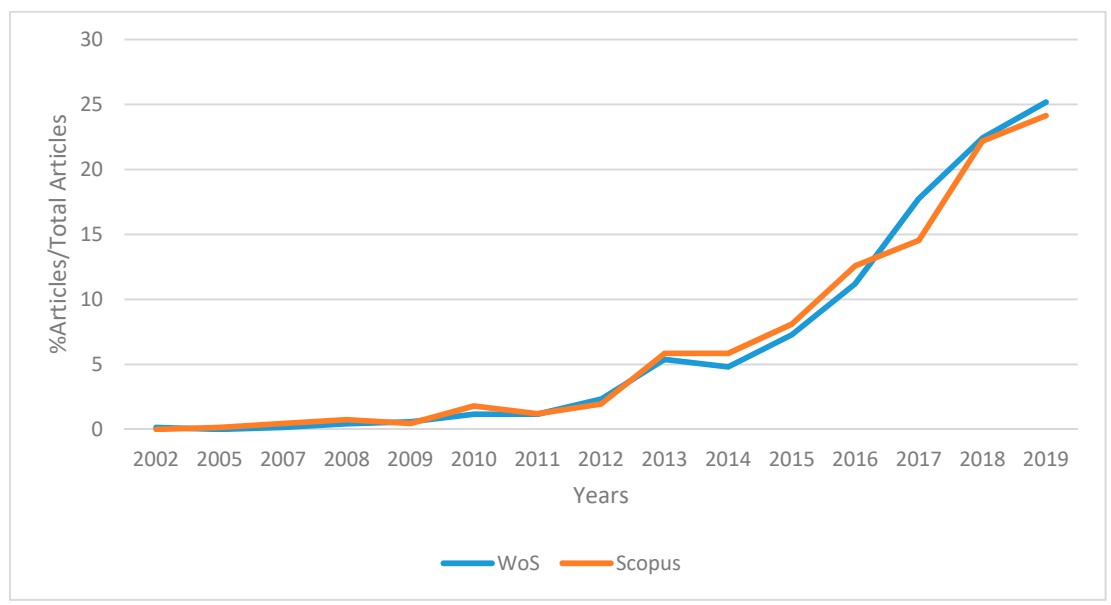

**Figure 4.** Evolution in the Number of Published Articles. Source: Own elaboration with Web of Science and Scopus data (2019).

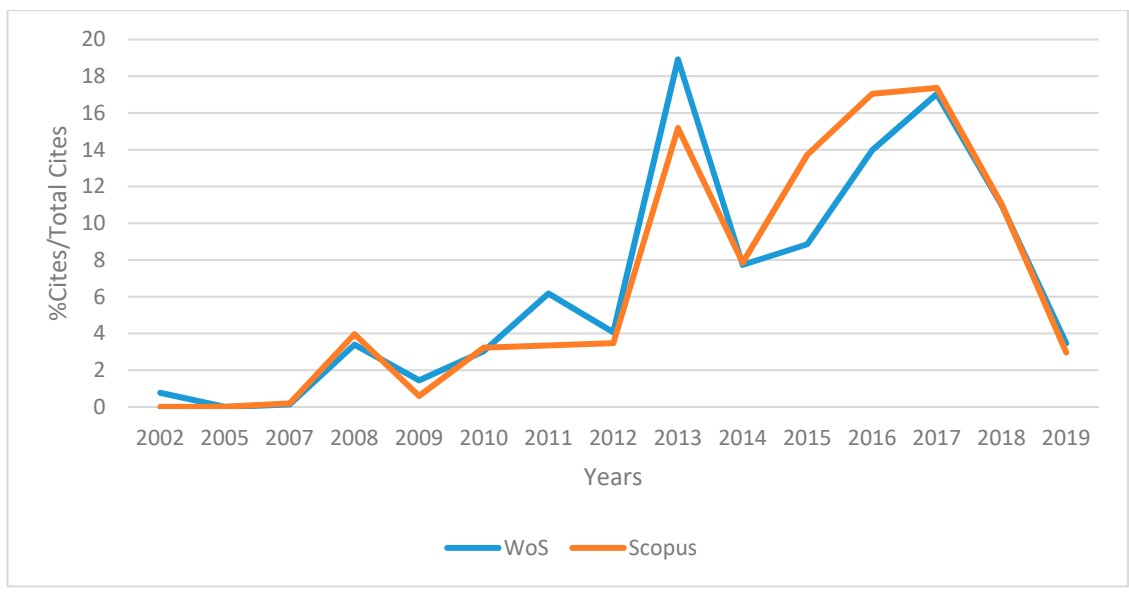

**Figure 5.** Evolution in Number of Citations. Source: Own elaboration with Web of Science and Scopus data (2019).

*3.2. Distribution by Knowledge Area*

There is a certain degree of diversity in the distribution by areas of knowledge, the most common being those related to environmental sciences, social sciences, engineering and ecology, among others (Table 5). This result confirms that the term GI is a broad and interdisciplinary concept, capable of responding to both small and large-scale geographical issues, and by both public managers, researchers and the general public [19]. In this sense, the literature review [27] on the relationships between the concepts of GI, ecosystem health, and human health and well-being already underlined the importance of the multidisciplinary character of this term, as well as the framework in which it must be analyzed.

**Table 4.** Ten most productive articles.

| No. | Authors | Title | Journal | Country/Institution | TC (Average WoS/Scopus) |
|---|---|---|---|---|---|
| **1** | [69] | Classifying and valuing ecosystem services for urban planning | Ecological Economics | Spain/Univ. Auton.Barcelona and Auton.Univ.Madrid | 483 |
| **2** | [70] | Coupling biogeochemical cycles in urban environments: ecosystem services, green solutions, and misconceptions | Frontiers in Ecology and the Environment | USA/Univ.California, Univ. Louisville, US Forest Serv. and Cornell Univ. | 386 |
| **3** | [71] | Urban gardens, agriculture, and water management: Sources of resilience for long-term food security in cities | Ecological Economics | Sweden/Stockholm Univ. | 143 |
| **4** | [72] | Watering our cities: The capacity for Water Sensitive Urban Design to support urban cooling and improve human thermal comfort in the Australian context | Progress in Physical Geography-Earth and Environment | Australia/Monash Univ. and Belgium/Katholieke Univ. Leuven | 130 |
| **5** | [73] | A watershed-scale design optimization model for stormwater best management practices | Environmental Modelling & Software | USA/US EPA and Tetra Tech Inc. | 130 |
| **6** | [74] | Characterizing the urban environment of UK cities and towns: A template for landscape planning | Landscape and Urban Planning | UK/Univ. Manchester | 115 |
| **7** | [75] | A rainwater harvesting system reliability model based on nonparametric stochastic rainfall generator | Journal of Hydrology | USA/Drexel Univ. and Columbia Univ. | 103 |
| **8** | [76] | Utilizing green and blue space to mitigate urban heat island intensity | Science of the Total Environment | UK/Univ. Bath | 90 |
| **9** | [77] | Permeable pavement as a hydraulic and filtration interface for urban drainage | Journal of Irrigation and Drainage Engineering | USA/Univ. Florida, Louisiana State Univ. and Italy/Bari Polytech Univ. | 84 |
| **10** | [78] | Life cycle implications of urban green infrastructure | Environmental Pollution | USA/Drexel Univ. | 81 |

Source: Own elaboration with Web of Science and Scopus data (2019). Note: TC: Total Cites.

**Table 5.** Distribution by Knowledge Area.

| Research Areas WoS | Articles | Research Areas Scopus | Articles |
|---|---|---|---|
| Environmental Sciences | 404 | Environmental Sciences | 552 |
| Water Resources | 205 | Social Sciences | 200 |
| Ecology | 101 | Agricultural and Biological Sciences | 128 |
| Engineering | 99 | Engineering | 105 |

Source: Own elaboration with Web of Science and Scopus data (2019).

### 3.3. Distribution by Institution

The US Environmental Protection Agency is the institution with the largest number of articles published in both databases. In WoS, the second and third position is occupied by the University of California System and the State University System of Florida, while in Scopus, these positions are occupied by Villanova University and Chinese Academy of Sciences (Table 6). The majority of the most influential institutions in this field of research are located in the United States, followed by China.

**Table 6.** Distribution of Articles by Institution.

| Institution | Articles | | Total Cites | | TC/A. | | H-Index | |
|---|---|---|---|---|---|---|---|---|
| | W | S | W | S | W | S | W | S |
| US Environmental Protection Agency | 44 | 41 | 732 | 1138 | 16.64 | 27.76 | 15 | 16 |
| University of California System | 22 | - | 588 | - | 26.73 | - | 8 | - |
| State University System of Florida | 21 | 9 | 595 | 269 | 28.33 | 29.89 | 8 | 6 |
| Drexel University | 18 | 12 | 319 | 348 | 17.72 | 29 | 5 | 6 |
| US Department of Agriculture | 17 | 3 | 504 | 3 | 29.65 | 1 | 7 | 1 |
| Villanova University | 17 | 16 | 261 | 325 | 15.35 | 20.31 | 8 | 10 |
| Chinese Academy of Sciences | 16 | 17 | 251 | 305 | 15.69 | 17.94 | 9 | 10 |

Source: Own elaboration with Web of Science and Scopus data (2019). Notes: W: WoS; S: Scopus.

The US Environmental Protection Agency is an organization dedicated to protecting the environment and human health. The University of California System is one of the most relevant institutions of higher education in the world, being made up of ten branches throughout the state. It stands out for its training and research work in biotechnology, computer science, environmental science, art and architecture.On the other hand, the State University System of Florida is a system of higher education centers that stands out in areas such as environmental science, engineering and economics and business. For its part, Villanova University is an American training and research center that stands out for its academic offering in areas such as law and economics and business. Finally, the Chinese Academy of Science is an institution that stands out for its contribution to the field of environmental protection and human health.

### 3.4. Distribution by Author

The distribution by authors shows that the largest number of documents published on this subject are by Montalto & Garg (each with 11 articles) in WoS, and Shuster (with 16 articles) in Scopus; these authors are followed by researchers such as Borst (Table 7). These authors have furthered their research career in this branch of knowledge in the second decade of the 21st century. Montalto's most cited article, "A rainwater harvesting system reliability model based on nonparametric stochastic rainfall generator" [75] (102 citations), assesses the viability of rainwater for residential uses through a rainwater harvesting model. Shuster presents as his most cited article "The role of trees in urban stormwater

management" [79] (83 citations). This article is in line with the previously mentioned article by Montalto and analyzes the role of trees in stormwater management. Garg's most cited work is titled "A new computational approach for estimation of wilting point for green infrastructure" [80] (51 citations), which proposes the development of a wilting point model based on the optimization approach of genetic programming in plant transpiration processes. Finally, Borst's most cited article, "Evaluation of Surface Infiltration Testing Procedures in Permeable Pavement Systems" [81] (25 citations), analyzes which surfaces are the most suitable for developing permeable pavements.

**Table 7.** Distribution by Author.

| Authors | Articles | | Total Cites | | TC/A. | | H-Index | | 1st Article | Last Article |
|---|---|---|---|---|---|---|---|---|---|---|
| | W | S | W | S | W | S | W | S | | |
| **Garg, A.** | 11 | 8 | 130 | 45 | 11.82 | 5.63 | 6 | 4 | 2017 | 2019 |
| Montalto, F. | 11 | 12 | 217 | 348 | 19.73 | 29 | 5 | 6 | 2010 | 2019 |
| Borst, M. | 10 | 10 | 98 | 137 | 9.8 | 13.7 | 6 | 7 | 2013 | 2019 |
| Shuster, W.D. | 9 | 16 | 118 | 722 | 13.11 | 45.13 | 6 | 10 | 2012 | 2019 |
| Brown, R.A. | 8 | 8 | 90 | 126 | 11.25 | 15.75 | 5 | 6 | 2013 | 2016 |
| Engel, B.A. | 8 | 4 | 83 | 78 | 10.38 | 19.5 | 5 | 4 | 2016 | 2019 |
| Liu, Y.Z. | 8 | 4 | 83 | 78 | 10.38 | 19.5 | 5 | 4 | 2016 | 2019 |
| Deletic, A. | 7 | 6 | 99 | 95 | 14.14 | 15.83 | 5 | 5 | 2017 | 2018 |
| Gadi, V.K. | 7 | 6 | 40 | 23 | 5.71 | 3.83 | 3 | 3 | 2017 | 2019 |
| Traver, R.G. | 7 | 8 | 161 | 259 | 23 | 32.38 | 6 | 7 | 2008 | 2019 |

Notes: W: WoS; S: Scopus. Source: Own elaboration with Web of Science and Scopus data (2019).

If the results are analyzed from the perspective of collaboration between authors, and taking into account those authors who have published a minimum of five research articles and 65 citations per document [82], it is observed that only six meet this condition (Figure 6). These are distributed in three clusters, Lee, J.G. and Shuster, W.D. being the most influential in terms of citations received and documents published.

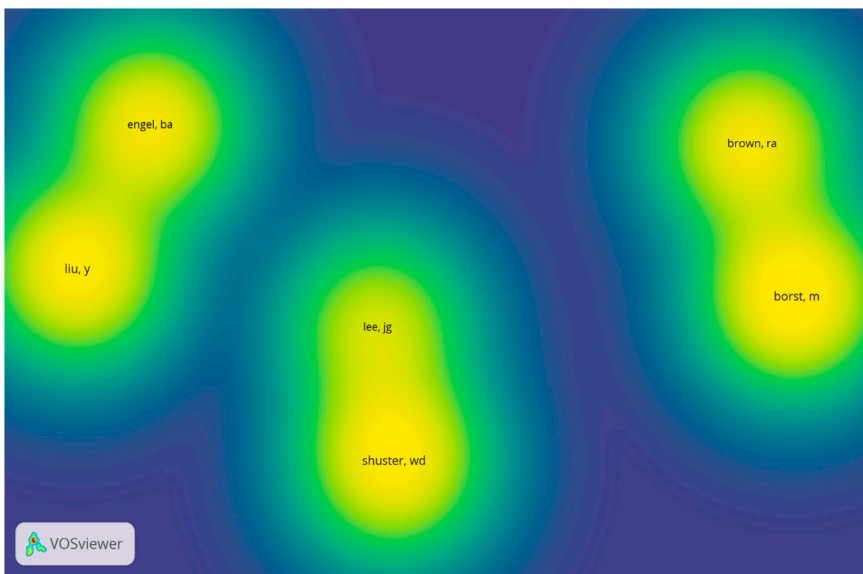

**Figure 6.** Density map of co-authorship network. Source: Own elaboration with Web of Science and Scopus data (2019) processed with VOSviewer software, developed by the Centre for Science and Technology Studies, Leiden University, The Netherlands.

### 3.5. Distribution by Journal

The distribution by journals shows that, in WoS, *Sustainability* has published the largest number of articles (33 articles) on GI and water, while in Scopus, the journal *Water* leads this ranking with 32 articles (Figure 7). The first of these journals, *Sustainability* promotes research in the environmental, social, cultural and economic sustainability of human beings, focusing on areas such as climate change, sustainable education or the creation of sustainability assessment tools.On the other hand, the journal *Water* focuses on water technology, governance and management. For example, it covers topics such as hydrology, water scarcity and flood risk. Therefore, the common ground of these journals is the concept of sustainability of water resources through GI as one of the fundamental aspects in managing the current situation of progressive climate change. They are followed in the ranking, in the case of WoS, by *Science of the Total Environment, Landscape and Urban Planning and Urban Forestry & Urban Greening*. In the case of Scopus, the following journals can be highlighted: *Science of the Total Environment*, *Ecological Engineering* and *Journal of Environmental Management*. In fact, if these data are supplemented with Table 4, it can be seen that *Science of the Total Environment* also includes one of the most frequently cited articles. It must be emphasized that the journals that publish the greatest number of articles on the importance of water in the operation of the GI are indexed in the first two quartiles of both databases, a guarantee of the quality of their editorial production. Furthermore, the journals previously mentioned in this section are also considered to be the most influential in terms of citations (Figure 8).

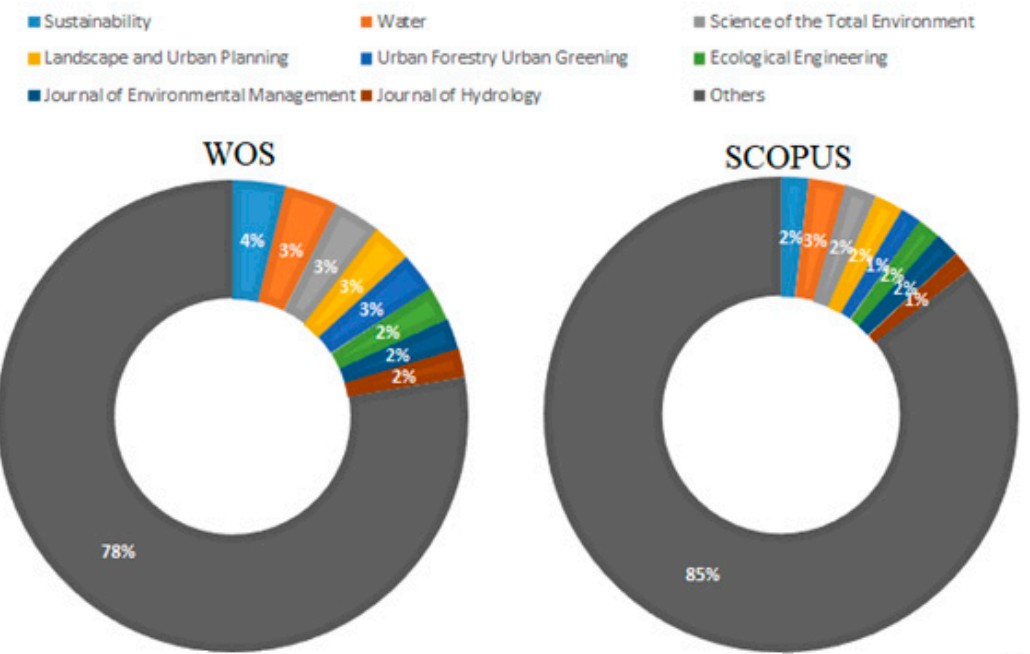

**Figure 7.** Distribution by most publishing journals. Source: Own elaboration with Web of Science and Scopus data (2019).

### 3.6. Distribution by Country and Language

The distribution by country shows that the United States, China, the United Kingdom and Australia are the most relevant countries in terms of publications in this scientific field, both in WoS and Scopus (Figure 9). They are followed by Italy, Germany and Canada, coinciding with the results obtained in the distribution by institutions. However, when adjusting the results according to population of each country, it is noted that Denmark and Australia are the most productive countries (Figure 10). As expected, the results of the distribution of articles by language confirm those obtained on the distribution by institutions, with English being the most predominant language, far above the rest.

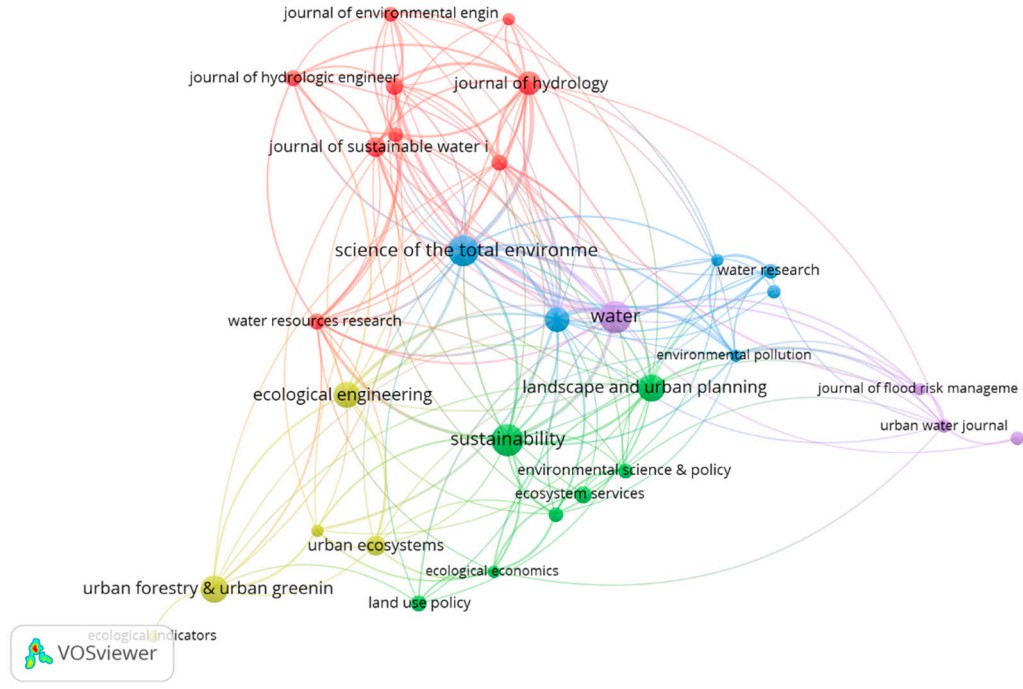

**Figure 8.** Science map of sources. Source: Own elaboration with Web of Science and Scopus data (2019) processed with VOSviewer software.

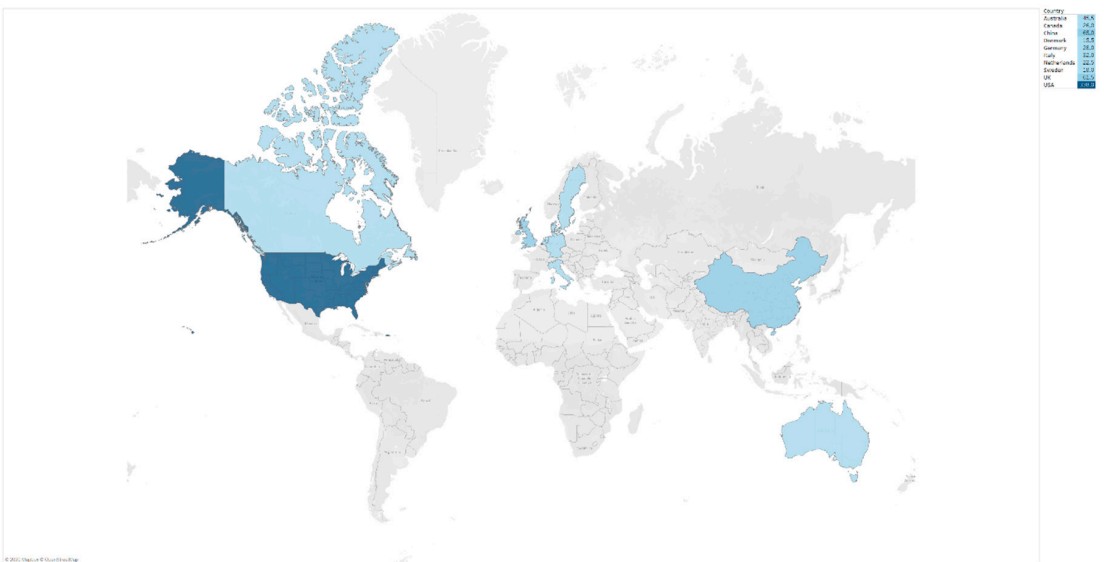

**Figure 9.** Distribution by Country. Note: The countries colored in blue are those with published articles. The darker the shade of that color, the greater the number of articles published. Source: Own elaboration with Web of Science and Scopus data (2019).

## 3.7. Analysis of Keywords and Latest Trends

In the analysis of the keywords, in the whole period from 2002 to 2019, the most prevalent concept in this type of research related to the importance of water in GI corresponds to that of *ecosystem services* (Table 8), which is defined as the conditions and processes in those ecosystems that generate and support human life [83].

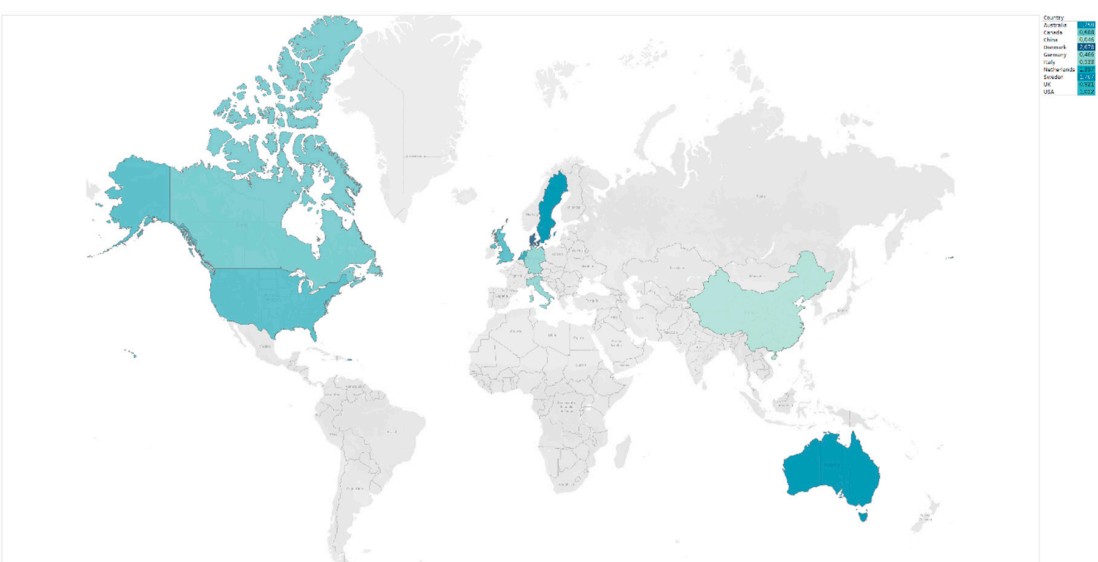

**Figure 10.** Distribution by Country adjusted by population. Note: The countries colored in blue are those with published articles. The darker the shade of that color, the greater the number of articles published. Source: Own elaboration with Web of Science and Scopus data (2019).

Gomez-Baggethun & Barton [69] indicate the suitability of the GI in the operation of ecosystem services to improve resilience and quality of life in urban cities, also mentioning the great socio-economic losses that would occur in the absence of ecosystems. Along these same lines, La Rosa & Privitera [84] propose the use of undeveloped areas as places of empowerment of ecosystem services through GI against a process of extensive and indiscriminate urbanization. Furthermore, Voskamp & Van de Ven [85] reinforce the role of ecosystem services as softeners of extreme weather conditions in urban environments, emphasizing the management of GI. Taken together, the importance of the concept of ecosystem services seems to demonstrate that the scientific community, within the wide range of existing definitions of this term, accepts to a greater extent those approaches that consider GI as a natural network of green and blue spaces that provides a wide variety of environmental services that contribute to human development and well-being [24,25,31].

In terms of most prevalent keywords, the above is followed in the ranking by the concept of management, more specifically, *stormwater management*. Keeley et al. [86] propose the use of GI to achieve the stormwater management objectives of urban revitalization and economic recovery. Part of the process of stormwater management, an effective and efficient way to reduce flood risks in urban areas, is the use of trees as GI [79]. Closely related to the concept of stormwater management is that of *Low Impact Development (LID)*, which consists of the use of GI in stormwater management processes in order to improve water quality [87]. Along with these terms there are also others of a more generic nature such as *runoff*, *performance*, *urban* and *quality*.

Nevertheless, it should be noted that *climate change* and *sustainability* are among the most widely used terms in the specialized literature, which shows the growing interest of the research community in planning GI solutions to guarantee sustainability in the global context of climate change. In fact, taking a look at the distribution based on Keywords Plus, *climate change* is the second most used. Therefore, from this perspective it is understood that the scientific community sees the use of GI as a tool or strategy to adapt spaces (especially those of an urban nature) to the new conditions imposed by climate change in order to mitigate its harmful effects [20].

**Table 8.** Most used keywords.

| 2002–2019 | | | 2002–2013 | | | 2014–2019 | | |
|---|---|---|---|---|---|---|---|---|
| Keywords | A | % | Keywords | A | % | Keywords | A | % |
| **Green infrastructure** | 382 | 12.37 | Green infrastructure | 39 | 7.69 | Green infrastructure | 343 | 11.95 |
| **Ecosystem services** | 115 | 3.72 | Water | 10 | 1.97 | Ecosystem services | 109 | 3.80 |
| **Management** | 97 | 3.14 | Stormwater management | 9 | 1.78 | Management | 92 | 3.20 |
| **Stormwater management** | 87 | 2.82 | Climate change | 8 | 1.58 | Runoff | 78 | 2.72 |
| **Runoff** | 84 | 2.72 | Energy | 8 | 1.58 | Stormwater management | 78 | 2.72 |
| **Water** | 84 | 2.72 | Low impact development | 8 | 1.58 | Water | 75 | 2.61 |
| **Climate change** | 74 | 2.40 | Runoff | 8 | 1.58 | Performance | 69 | 2.40 |
| **Performance** | 74 | 2.40 | Stormwater | 7 | 1.38 | Stormwater | 66 | 2.30 |
| **Stormwater** | 73 | 2.36 | Sustainability | 7 | 1.38 | Climate change | 65 | 2.26 |
| **Low impact development** | 67 | 2.17 | Biodiversity | 6 | 1.18 | Low impact development | 59 | 2.06 |
| **Urban** | 62 | 2.01 | Reuse | 6 | 1.18 | Urban | 58 | 2.02 |
| **Urbanization** | 52 | 1.68 | Ecosystem services | 5 | 0.99 | Urbanization | 48 | 1.67 |
| **Quality** | 50 | 1.62 | Management | 5 | 0.99 | Quality | 47 | 1.64 |
| **Sustainability** | 48 | 1.55 | Performance | 5 | 0.99 | Systems | 46 | 1.60 |
| **Systems** | 48 | 1.55 | Vegetation | 5 | 0.99 | Impact | 43 | 1.50 |
| **Impact** | 46 | 1.49 | Water reuse | 5 | 0.99 | Sustainability | 42 | 1.46 |
| **Cities** | 45 | 1.46 | City | 4 | 0.79 | Bioretention | 41 | 1.43 |
| **Bioretention** | 44 | 1.42 | Climate | 4 | 0.79 | Cities | 41 | 1.43 |
| **Design** | 44 | 1.42 | Design | 4 | 0.79 | Design | 40 | 1.39 |
| **Biodiversity** | 43 | 1.39 | Hydrology | 4 | 0.79 | Model | 40 | 1.39 |

Source: Own elaboration with Web of Science and Scopus data (2019).

The evolution of the number of published articles represented in Figure 4 shows that the year 2013 is the turning point from which the number of publications in which GI is related to the water cycle increases. A comparison of the most commonly used keywords before and after that year shows that, in the first subperiod (2002–2013), the relationship between both concepts materialized in terms such as stormwater management, climate-change or energy, among others. Therefore, in this first period the focus of the scientific community seems to be on terms related to new approaches to engineering, urbanism and land use planning to achieve a better integration between the urbanized space and the natural environment in terms of reducing and controlling the amount and energy of rainwater and better adapting to the phenomena of Climate Change. In the second period, when the number of publications is increasing, the concept covers a wider field of action and the key term *ecosystem services* gains special relevance. This fact shows that, during this period, the concept covers a wider spectrum of scales, and the multifunctionality of GIs as a source of a wide variety of key environmental services for human well-being and as a tool for solving environmental problems through nature-based solutions takes on special relevance.On the other hand, it is in this period when in a great number of institutions globally the concept is assumed as a central element in their strategies of nature conservation, adaptation to Climate Change and urban sustainability.

Similarly to earlier comments on the analysis of Keywords Plus, despite the fact that climate-change and sustainability have fallen in the ranking of the most used keywords before and after the turning point year, the percentage in relation to the total number of keywords used is higher, which is in line with the previous reflection on a trend towards the search for strategies against climate change based on sustainable GI solutions.

However, at the beginning of the study period it was observed that neither green infrastructures nor water were found in the key words of the works published in the 2002–2005 period (Figure 11), and they were only mentioned in the concept of sustainability and best management practices (BMPs).

Furthermore, Figure 12 represents a cluster analysis of all keywords in specific time periods, in this case by yearly intervals from 2013 (the moment at which the great increase in the number of publications occurs, as shown in Figure 4) to 2019. Hence, as can be seen in purple, the most frequently used in 2013 are *green infrastructure, stormwater management/runoff, urban, hydrology, health* and *benefits*. Thus, it is detected that, at the beginning of the period analyzed, one of the key aspects of GI is the management of rainwater for the benefit of the well-being of cities. In subsequent years, shown in green or greenish-yellow, new terms such as sustainability, rainwater harvesting, ecological design, water quality or microclimate begin to appear. Finally, in yellow, the most recent terms represent aspects such as air quality, urban trees, nature based solutions, adaptation or urban stormwater. In light of this, an evolution of the relationship between GI and water is made apparent which, starting from the concern for stormwater management, has reached a stage in which scientific awareness has increased in adapting urban environments to the requirements of environmental sustainability through natural solutions that not only improve the well-being of society and the quality of water, but also of the air.

In addition, a cluster analysis of the distribution of the keywords by topic is performed and the result shows a classification into five main groups, differentiated by color (Figure 13). The first cluster, shown in blue (both light and darker blue) refers to the relationship of GI with the collection and increase of water availability, and what tools can be articulated to carry out this task. A second cluster, shown in yellow, represents those concepts referring to the environmental quality of urban spaces that are related to GI. The red cluster revolves around concepts related to ecology, environmental services, and climate change adaptation and mitigation policies, which could be grouped within NbS. The green cluster revolves around the concept of rainwater and management of this source, as well as in the elements that enhance pavement permeability. Finally, the group in purple represents the concept of sustainability, linked to efficiency in the management of groundwater and reservoirs.

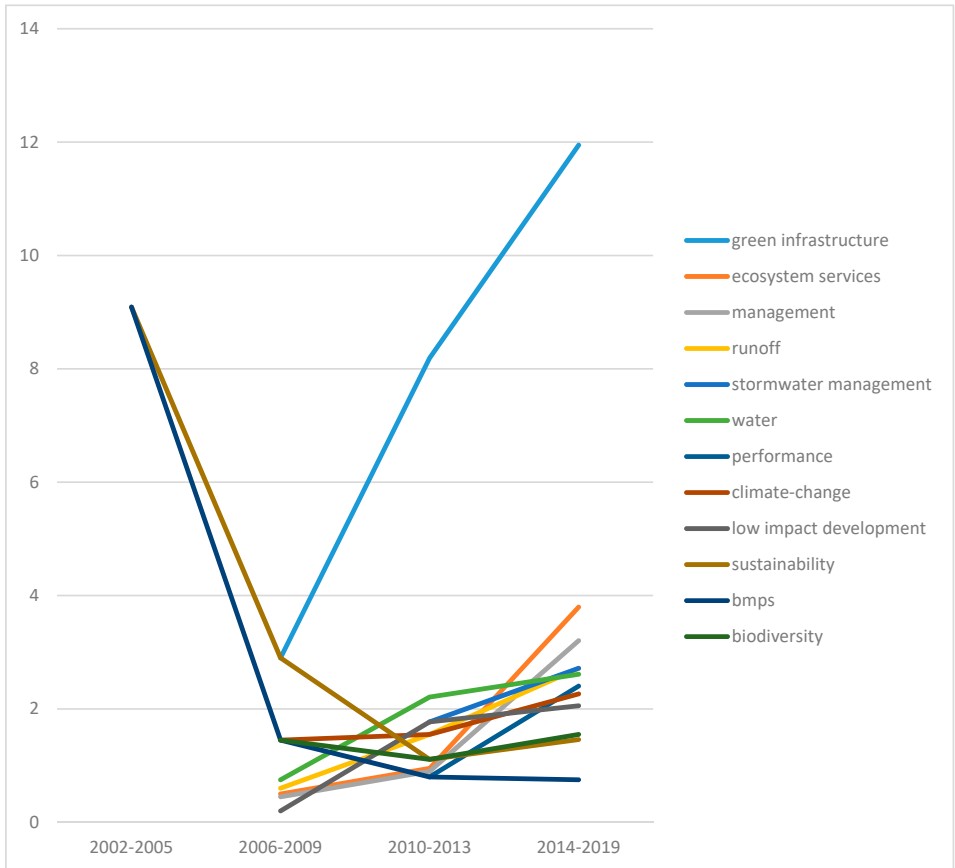

**Figure 11.** Normalized frequency of occurrence for each keyword among papers published in the time period considered. Source: Own elaboration with Web of Science and Scopus data (2019).

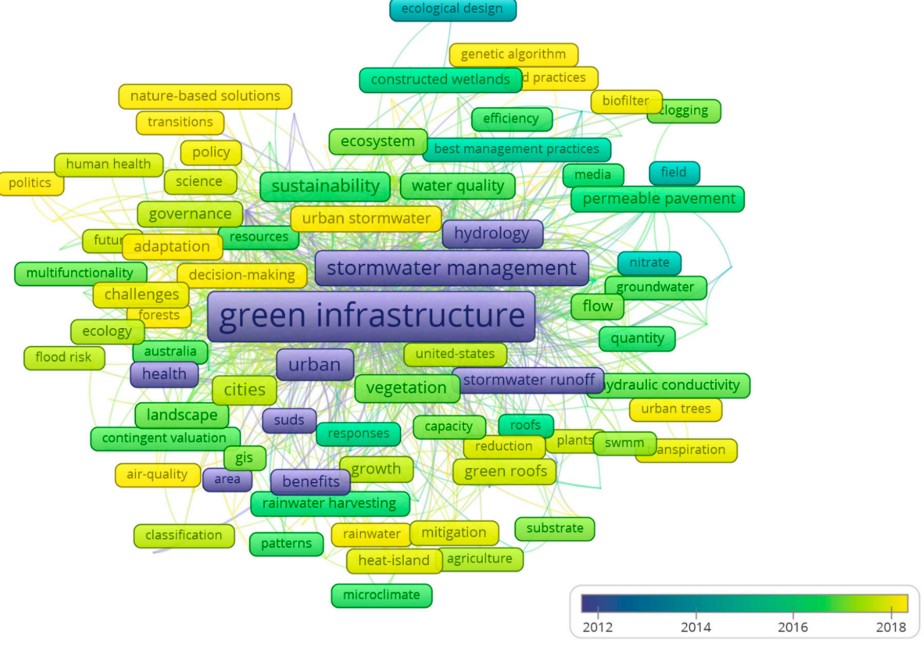

**Figure 12.** Keyword Cluster by Year. Source: Own elaboration with Web of Science and Scopus data (2019) processed with VOSviewer, developed by the Centre for Science and Technology Studies, Leiden University, The Netherlands.

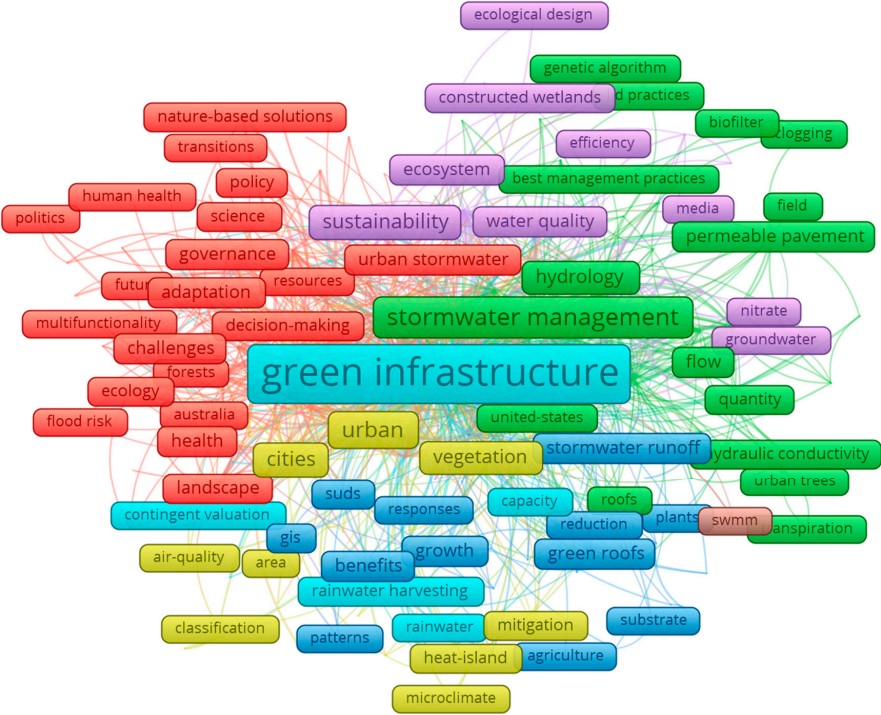

**Figure 13.** Keyword Cluster by Topic. Source: Own elaboration with Web of Science and Scopus data (2019) processed with VOSviewer software, developed by the Centre for Science and Technology Studies, Leiden University, The Netherlands.

This distribution in clusters becomes even more precise in three groups if we analyze the words used only in titles and abstracts of articles that deal with the relationship between green infrastructure and the water cycle (Figure 14). On the one hand, in red, the concepts of ecosystem service, sustainability, policy and challenges are highlighted, indicating that the sustainable management of ecosystem services is a challenge to be met by the authorities; in blue, nature based solutions such as trees, forests and other types of vegetation are represented, and in green, aspects related to stormwater management such as runoff or low impact development are found. Overall, the three fundamental pillars of the abstracts and article titles that relate green infrastructure to the water cycle are ecosystem services, nature based solutions and stormwater management.

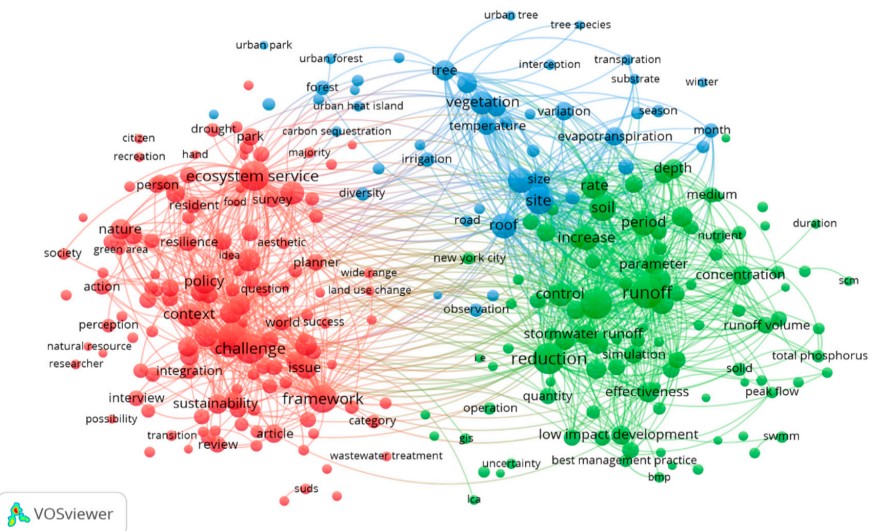

**Figure 14.** Cluster by topic of words in abstracts and titles of published articles. Source: Own elaboration with Web of Science and Scopus data (2019) processed with VOSviewer software, developed by the Centre for Science and Technology Studies, Leiden University, The Netherlands.

## 4. Conclusions

This work serves to confirm that, in the field of water study, the term GI is relatively new, since it first appeared in the early 21st century. The use of this term grew exponentially from 2013, coinciding with the increase in work and research related to the environmental services of ecosystems and the possibilities that nature offers to tackle problems related to urban planning, the management and purification of water, agri-food security, flood control or adaptation to climate change. In the case of Europe, it coincides with the GI strategy of the European Commission to improve the economic benefits of this tool and thus achieve its biodiversity objectives for 2020. This growth in the volume of publications on GI is expected to continue in the future, not only because of the economic benefits that GIs generate but also because of the ecological, social, and public health and welfare benefits.

On the other hand, it is worth mentioning that the most cited articles during the period studied, to a large extent, deal with issues related to the conservation and management of IGs for the improvement of the environmental quality of urban environments, which shows a growing involvement of urban planning professionals in this field.

The non-existence of a universally accepted definition of GI calls for further efforts in this type of literature that favor consistency and clarity in this field of research. Despite this, it appears from this analysis that we are dealing with a field of research widely spread across all continents (with the exception of Africa), with the United States, China, United Kingdom and Australia publishing the greatest number of articles on GI and water. Likewise, its multidisciplinary character is reflected in the diversity of areas of knowledge with which it interacts, including topics such as those related to protection against natural disasters, the provision and regulation of water resources such as groundwater and reservoirs, the planning and environmental improvement of urban spaces, health, the ecology and conservation of biodiversity, and adaptation to the effects of climate change.

This research indicates that in the scientific literature related to GI and water management the concept with greatest relevance is ecosystem services, a fact that values the capacity of these infrastructures to facilitate multiple goods and services related to the water cycle, such as the potable water supply, climatic regulation, flood control, water purification or disposition of spaces of recreation for the population.

The scientific literature focusing on the importance of GI in water management has been dominated by concepts such as ecosystem services, stormwater management, climate change and sustainability, terms which are considered to be the most widely used in this field. In fact, the importance of the concept of ecosystem services indicates that green infrastructure is an essential tool for improving human well-being by NbS. Also, the latest keyword trends focus on aspects related to NbS such as stormwater management, forests and green rooves, among others. Therefore, from this perspective, the opportunity is opened for new lines of research based on the analysis of other elements related to GI, such as groundwater.

On the other hand, GI offers the opportunity to lessen the adverse effects of climate change and, thus, generate important benefits from the perspective of human health and well-being. In the current context of the global health crisis, another recommendation for future research is to continue advancing precisely the analysis of the role of GI from the perspective of public health. Consequently, it is also necessary to improve the definition of GI and future work is needed to advance the search for new methodologies that facilitate an assessment of the monetary and non-monetary benefits of GI.

Likewise, the current global crisis opens up lines of research in the field of green infrastructure as an emerging sector in the process of economic reconstruction after the harmful consequences of the Covid-19 pandemic.

This work is not exempt from certain limitations as it has been limited to an analysis of the role of GI in water management. It would be interesting to repeat this analysis process for other types of infrastructure of a similar nature. Likewise, another proposal for a future line of research consists of not limiting the analysis to water resources but rather extending it to other natural resources.

**Author Contributions:** Conceptualization, J.L.C.-M., J.M.-G., N.R.-L., and J.d.P.-V.; Methodology, J.L.C.-M., J.M.-G., N.R.-L., and J.d.P.-V.; Investigation, J.L.C.-M., J.M.-G., N.R.-L., and J.d.P.-V.; Writing—Review and Editing, J.L.C.-M., J.M.-G., N.R.-L., and J.d.P.-V. All authors have read and agreed to the published version of the manuscript.

**Funding:** This research received no external funding.

**Conflicts of Interest:** The authors declare no conflict of interest.

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
