# Peer review of "Green Infrastructure and Water: An Analysis of Global Research"

_water, doi:10.3390/w12061760_

Round 1
Reviewer 1 Report
I was able to reread the article, already of excellent quality, after the suggestions I had reported in the previous review. All the errors have been corrected.
Author Response
Thank you very much for your comments, we really appreciate that you enjoy our paper. We are sure that your suggestions have increased the quality of the paper.
Reviewer 2 Report
Authors should reconsider how they will reorganised text of the article to fill gap form line 87 to line 89 (2/3th of the page).
Authors should reconsider how they will reorganised text of the article to fill gap form line 87 to line 89 (2/3th of the page).
Also, in the lines 482-484 is one sentence in Spanish language. Should be translated to English or deleted.
Author Response
Thank you very much for your comments. We have addresed the issue regarding the gap in lines 87-89, which we have deleted it. Also, we have translated the sentence in Spanish, which has been highlighted in yellow in lines 478-480.
Reviewer 3 Report
- The aim and method
The aim of the article is clear as well as the research method (based on the bibliometric analysis) description.
- The structure of the article
The structure of the article is correct. The main emphasis was placed on the description of the results.
The Introduction, which is well composed , presents the subject of research effectively. I would only suggest considering whether the problem of incorporating water into the definition of GI depends only on projects or research (see lines 78-81) or if instead we are dealing with a separate approach to GI, with a strong emphasis on water management issues (including rainwater management). Sometimes this is revealed in a change of name to blue-green infrastructure.
- Results and conclusions
Their range closely relates to the possibilities of the tool used. Among the issues analysed, the most interesting for readers might be the subsection on Analysis of Keywords and Latest Trends. The results of the research presented here can be directly useful for people who want to present their own research and are looking for their factual background.
- Summing up
The article is suitable for publication, almost unchanged. I would only suggest considering the remark presented in point 2) and removing some minor errors:
line 112: there should be GI instead of IG
lines 482-484: the text is in Spanish.
I have no doubt that the article will be often cited.
Author Response
We are glad to know that you have enjoyed the last version of our paper. We really appreciate the comments you have made regarding the introduction, in which we have included in lines 79-82 (highlighted in yellow), as well as the correction in line 108 (GI instead of IG) and the translation of the text in Spanish (lines 478-480, highlighted in yellow). Thank you very much for your comments, they encourage us to keep working and the quality of the paper has increased significantly.
This manuscript is a resubmission of an earlier submission. The following is a list of the peer review reports and author responses from that submission.
Round 1
Reviewer 1 Report
The paper is very well written. It is innovative in the fields of study related to green infrastructures and water management. In my opinion, this work is interesting for all researchers in these fields. I found only a typing error at the line 15: repetition of "trends".
Reviewer 2 Report
The subject of the article is interesting but there are some aspects relating to the different types of urban green infrastructures and how these systems function that should be added.
Please reconsider the abstract form.
Line 112 the image is too large. Advantages: natural design, sustainability, ecological effect. Some of the green infrastructures like: like green walls, green roofs, bioswales, bioretention system etc have also weaknesses related to cost maintenance.
Also the other images used in the article are too large.
Which is the aim of the study? The high interest of this subject in the researches from the last years? The correlation between GI and water management and highlight the most cited articles? Please be more specific
On the other hand it is clear that the research in this field is a current subject and in the future there will be more articles, so please formulate relevant conclusion which emerges from the research.
Best regards,
Reviewer.
Reviewer 3 Report
The paper proposes an interesting topic related with GI and water. However, the limited literature that are considered favors a misunderstood of the concepts. The authors consider a interpretation of the concept from Clean Water Act (that it is more a definition of NbSs) and neglect the other approaches (e.g. Pauleit, Haase, Formann, Laffortezza, Andersson). The GI is not related only with the water, but also by the GREEN. For Water is Blue Infrastructure, a different concept that is focused on aquatic ecosystems. It has a huge social dimensions that is completely neglected in the paper. The cultural ecosystem service of blue infrastructure have to be considered (Hossu et al, 2019, Ecosystem Services). Also, the GI is more oriented on the planning process (Hansen and Pauleit, 2014, Ambio). NbSs are a form to implement GI (not a synonymous).
To consider the importance in the field, it is not important only the number of the paper, but also the influence of these papers. Biosciences, Ambio, Science for Total Environment do not have many papers, because of the journal policy (unconsidered in these analysis), but the quality is very high. And that do not means only citations.
The figures need consistent revisions. For example, Figure 1 consider in Health and wellbeing: biodiversity conservation, water purification, groundwater recharge. There are regulatory ecosystem services.